# An Inducible Diabetes Mellitus Murine Model Based on MafB Conditional Knockout under MafA-Deficient Condition

**DOI:** 10.3390/ijms21165606

**Published:** 2020-08-05

**Authors:** Zhaobin Deng, Yuka Matsumoto, Akihiro Kuno, Masami Ojima, Gulibaikelamu Xiafukaiti, Satoru Takahashi

**Affiliations:** 1Department of Anatomy and Embryology, Faculty of Medicine, University of Tsukuba, 1-1-1 Tennodai, Tsukuba, Ibaraki 305-8575, Japan; s1836036@s.tsukuba.ac.jp (Z.D.); s1611817@u.tsukuba.ac.jp (Y.M.); s1536043@u.tsukuba.ac.jp (G.X.); 2School of Comprehensive Human Sciences, Doctoral Program in Biomedical Sciences, University of Tsukuba, 1-1-1 Tennodai, Tsukuba, Ibaraki 305-8575, Japan; 3School of Medical Science, University of Tsukuba, 1-1-1 Tennodai, Tsukuba, Ibaraki 305-8575, Japan; 4PhD Program in Human Biology, School of Integrative and Global Majors, University of Tsukuba, 1-1-1 Tennodai, Tsukuba, Ibaraki 305-8575, Japan; 5Laboratory Animal Resource Center (LARC), University of Tsukuba, 1-1-1 Tennodai, Tsukuba, Ibaraki 305-8575, Japan; mk03j341@md.tsukuba.ac.jp; 6Life Science Center, Tsukuba Advanced Research Alliance (TARA), University of Tsukuba, 1-1-1 Tennodai, Tsukuba, Ibaraki 305-8575, Japan; 7International Institute for Integrative Sleep Medicine (WPI-IIIS), University of Tsukuba, 1-1-1 Tennodai, Tsukuba, Ibaraki 305-8575, Japan; 8Transborder Medical Research Center, Faculty of Medicine, University of Tsukuba, 1-1-1 Tennodai, Tsukuba, Ibaraki 305-8575, Japan

**Keywords:** diabetes mellitus, murine model, MAFA, MAFB

## Abstract

Diabetes mellitus is an increasingly severe chronic metabolic disease that is occurring at an alarming rate worldwide. Various diabetic models, including non-obese diabetic mice and chemically induced diabetic models, are used to characterize and explore the mechanism of the disease’s pathophysiology, in hopes of detecting and identifying novel potential therapeutic targets. However, this is accompanied by disadvantages, such as specific conditions for maintaining the incidence, nonstable hyperglycemia induction, and potential toxicity to other organs. Murine MAFA and MAFB, two closely-linked islet-enriched transcription factors, play fundamental roles in glucose sensing and insulin secretion, and maintenance of pancreatic β-cell, respectively, which are highly homologous to human protein orthologs. Herein, to induce the diabetes mellitus model at a specific time point, we generated *Pdx1*-dependent *Mafb*-deletion mice under *Mafa* knockout condition (*A0B^Δpanc^*), via tamoxifen-inducible Cre-loxP system. After 16 weeks, metabolic phenotypes were characterized by intraperitoneal glucose tolerance test (IPGTT), urine glucose test, and metabolic parameters analysis. The results indicated that male *A0B^Δpanc^* mice had obvious impaired glucose tolerance, and high urine glucose level. Furthermore, obvious renal lesions, impaired islet structure and decreased proportion of insulin positive cells were observed. Collectively, our results indicate that *A0B^Δpanc^* mice can be an efficient inducible model for diabetes research.

## 1. Introduction

Diabetes mellitus (DM), a complex and chronic metabolic disease, has been one of the most critical public health problems over the last century, because of its high mortality and morbidity [1,2]. The number of people afflicted with DM has quadrupled during the past 30 years, reaching an epidemic proportion at an alarming rate [3]. Furthermore, it is also one of the top ten causes of death. Owing to chronic hyperglycemia, several complications such as nephropathy, neuropathy, and retinopathy arise, further exacerbating the DM landscape. Clinically, two of the most common types are type 1 diabetes (T1D), and type 2 diabetes (T2D), which are associated with the destruction of insulin-producing pancreatic β-cells and relative insulin deficiency caused by dysregulation of β-cells, respectively [4]. Owing to limited understanding of the pancreatic β-cell dysfunction mechanisms, management of the disease mainly relies on external insulin administration, which is costly and time-consuming. Therefore, in-depth understanding of the mechanisms for pancreatic β-cell dysfunction is urgently required, which is also the foundation for the development of novel therapeutic approaches.

An increasing amount of evidence have demonstrated that islet-enriched transcription factors play fundamental roles in regulating a series of β-cell events, including development, maintenance, and maturation. Among them, MAFA and MAFB, two counterparts exhibiting different expression patterns have attracted attention [5,6,7]. Particularly, in rodents, *Mafb* starts its transient production during the development of β-cells around embryonic day 10.5 (E10.5) and are then restricted to α-cells postnatally. In contrast, *Mafa* expression in β-cells initiates at E13.5 and its function persists into adulthood [8,9]. Many reports have uncovered the specific roles of *Mafa*, which is strongly associated with glucose-stimulated insulin secretion (GSIS) and maintenance of islet structure [9,10]. In contrast, *Mafb* knockout mice were restricted only in the embryonic phase but not the postnatal phase, owing to its embryonic lethality [11,12]. Recently, Conrad et al. [11] reported that pancreatic specific *Mafa* and *Mafb* double knockout mice showed significant hyperglycemia in parallel with less insulin positive cells. Meanwhile, in our previous study [13], we found that the mice that had a specific *Mafb* deletion in pancreatic β-cells under *Mafa*-deficient conditions were vulnerable to developing diabetes with high-fat diet (HFD) feeding. Collectively, this indicates that the deletion of both *Mafa* and *Mafb* are potent targets for the induction of DM.

Animal modeling, an efficient approach for exploring and characterizing disease etiology and pathophysiology, also provides novel insights into the development of therapeutic treatments [14,15,16,17]. Consequently, various diabetic animal models have emerged, such as chemically-induced diabetic models, non-obese diabetic (NOD) mice, Akita mice, and Bio-Breeding (BB) rats. Although these models are very useful and widely applied in diabetes research, there are several disadvantages, such as the fact that the Akita and streptozotocin (STZ)-induced diabetic mice rely on genetically or chemically disturbing the limited number of pathogenic factors of DM, which are of little relevance to clinical cases, and NOD mice need specific pathogen-free conditions to maintain the incidence [14]. Thus, models based on a deeper understanding of molecular mechanisms are urgently needed. Effective and reasonable approaches guarantee the success of animal models. During the past decades, the emergence of animal models modified by the tamoxifen-inducible Cre-loxP recombination system have provided cutting-edge insights into gene function. In particular, it prevents pre-adult lethality caused by prenatal or neonatal gene manipulation, to induce gene modification at desired time points.

In considering the unique roles of MAFA and MAFB in β-cell development and maturation, here, we aimed to develop a more efficient and inducible DM model.

## 2. Results

### 2.1. Validation of Cre Recombinase Efficiency in Pancreatic Islets Using R26GRR/Pdx1-CreER^TM^ Mice

A site-specific recombination system is a powerful tool for exploring gene functions by precisely manipulating gene expression in mice. Among which, the Cre-loxP system is widely used in monitoring the target gene expression temporally and spatially. Thus, the cutting efficiency of Cre recombinase guarantees the deletion of specific genes. To evaluate the Cre recombinase activity in pancreatic cells, we generated reporter mice by crossing *R26GRR* reporter mice with *Pdx1-CreER*^TM^ mice; the former exhibits different signals before (green) and after (red) Cre-mediated flox cassettes excision is activated. The genotype of the progeny was confirmed by polymerase chain reaction (PCR) with single and bright bands observed. Following tamoxifen (TAM) injection, both male and female mice showed predominant red fluorescence within islets; however, the background of Cre recombination was observed in male mice without TAM administration (Figure 1). Collectively, this indicates that the *Pdx1-CreER*^TM^ recombination system bore a relatively high Cre recombinase efficiency.

### 2.2. Assessment of Glucose Tolerance in A0B^Δpanc^ and A0B2 Mice

The body weights of the mice in each group were recorded from 4 weeks to 16 weeks post-TAM injection. The body mass of *A0B^Δpanc^* male mice was significantly lower than that in the *WT* group, while it was comparable between these two groups of female mice. Next, we evaluated the effects of inducible *Mafb* deletion in pancreatic progenitor cells under the *Mafa*-null knockout condition. The intraperitoneal glucose tolerance test (IPGTT) was implemented to assess the ability of glucose clearance among wild-type (*WT*), *A0B2* and *A0B^Δpanc^* mice, at designated time points, 0 weeks (before TAM injection), 4 weeks and 16 weeks post-TAM injection (Figure 2). After 4 months of breeding, the male mice in the *A0B^Δpanc^* group showed weaker glucose clearance ability, and glucose level peaked at around 800 mg/dL from 360 mg/dL after 30 min of glucose induction. Furthermore, the glucose levels in the *A0B^Δpanc^* group were markedly higher compared with the *A0B2* and *WT* groups at all time points during the following 90 min, before reducing to 590 ± 81 mg/dL (Figure 2C), which was comparable with the peak level of *A0B^Δpanc^* mice at 4 weeks. Interestingly, there were no striking differences between the *WT* and *A0B2* groups after a slight fluctuation of glucose levels at 2 h in weeks 4 and 16. In contrast, in female mice, the blood glucose levels showed similar patterns at 4 and 16 weeks. During the 2 h monitoring, the blood glucose levels increased rapidly and reached peak level at 30 min after glucose administration, then decreased to normal levels gradually (Figure 2E,F). The collective results indicated that the inducible deletion of *Maf**b* under the *Mafa* knockout condition impaired the glucose tolerance ability, causing hyperglycemia in *A0B^Δpanc^* mice, especially male mice.

### 2.3. Analysis of Metabolic Parameters in Urine and Blood

The parameters in urine provide effective evidence for various disease diagnoses, such as diabetes and urinary tract infection [18]. Thus, the urine glucose test was administrated by recruiting 16-week post-TAM mice. In male mice, the glucose level in the *A0B^Δpanc^* group was noticeably higher in comparison with the *WT* and *A0B2* groups, while, there was no considerable difference between these two groups (*A0B^Δpanc^* vs. *A0B2*: *p* = 0.000038; *A0B^Δpanc^* vs. *WT*: *p* = 0.000084; *A0B2* vs. *WT*: *p* = 1) (Figure 3A). In contrast, for female mice, the urine glucose concentrations had a different trend, and was comparable among the three groups (Figure 3B).

Among the many diabetic complications, diabetic nephropathy (DN) is probably one of the most detrimental, owing to its high morbidity and mortality. Therefore, clinical indicators, blood urea nitrogen (BUN) and serum creatinine (CRE) were also assayed. However, there were no statistical significances in the values of CRE and BUN in either male or female mice, indicating that the kidney function did not show obvious dysfunction at 16 weeks post-TAM injection (Figure 4).

### 2.4. Histological Changes in Kidneys of Male A0B^Δpanc^ Mice

Owing to the high blood and urinary glucose levels in male *A0B^Δpanc^* mice, renal histological analysis was performed to further detect possible pathological changes in the kidneys. Briefly, at 21 weeks, hematoxylin–eosin (H&E) and Masson trichrome (MT) staining were performed on male mice of indicated genotypes (Figure 5). Visually, the male mice in the *A0B^Δpanc^* group exhibited obvious pathological alterations compared with *WT* group, including mesangial matrix expansion and collagen deposition in mesangial matrix. Collectively, these data illustrate that there were moderate renal pathological changes in *A0B^Δpanc^* male mice.

### 2.5. Changes in the Histology of the Pancreas in A0B^Δpanc^ Mice

To analyze the impaired glucose tolerance in *A0B^Δpanc^* mice more comprehensively, immunohistochemistry (IHC) was performed in *WT*, *A0B2* and *A0B^Δpanc^* mice, by means of anti-insulin and anti-glucagon antibodies. Visually, the proportions of insulin positive cells in pancreatic islets were lower in both *A0B^Δpanc^* male and female mice compared with those in the *WT* group (Figure 6A,D). Furthermore, the numbers and percentages of insulin and glucagon positive cells were counted and calculated (Figure 6B,C,E,F). There was a marked reduction of the insulin positive cell number in both *A0B^Δpanc^* male and female mice. Interestingly, a significant difference was observed between male and female mice in the *A0B^Δpanc^* and *A0B2* groups, respectively.

In contrast, the amount of glucagon positive cells within the islets of *A0B^Δpanc^* mice were slightly increased compared with those of the *WT* mice, which was consistent with the IHC results. More specifically, the glucagon positive cell number was boosted in *A0B^Δpanc^* male mice compared with that in the *WT* group. However, the increase seen in *A0B^Δpanc^* female mice was milder. The architecture demolition in *A0B^Δpanc^* mice was therefore more obvious in *A0B^Δpanc^* male mice. While there is no obvious immune cell infiltration in pancreatic tissue of *A0B^Δpanc^* male mice.

## 3. Discussion

In 2015, the number of diabetes patients aged 20 to 79 years old reached 415 million. Under the current trajectory, the number of people in this age group suffering from diabetes will reach 642 million by 2040 [19]. Facing the dramatic increase in patients, the diabetes models are regarded as potent tools to elucidate the pathogenesis of human DM and develop novel therapeutic targets for treatment. To address this vital objective, various animal models that span across multiple species and strategies have been used. Significantly, small rodents, especially rats and mice, are important candidates for preclinical research on metabolic disorders [20,21]. In mammals, the physiological features of mice are closer to humans than those of other species. Moreover, the appropriate size and short reproductive cycle also drive the mice model at the forefront of DM research.

Here, we generated a novel inducible diabetic mouse model, while the specific pathological features were evaluated. Simply, the transcription factor *Mafb* was conditionally deleted in pancreatic progenitor cells under *Mafa* knockout condition (termed *A0B^Δpanc^* mice). *A0B^Δpanc^* mice displayed obvious hyperglycemia compared with the *A0B2* and *WT* mice, at least 2 h following glucose administration (Figure 2). In addition, the urine glucose level in *A0B^Δpanc^* mice presented a dramatic increase compared with the other two groups (Figure 3). To illustrate more precisely, pathological analysis was performed, and showed that the conditional knockout of *Mafb* mice produced irreversible damage and reduction in islet structure and insulin positive cell numbers in both male and female mice (Figure 6A–E). Intriguingly, the proportions of insulin positive cells of indicated genotypes showed different patterns in male and female mice. Moreover, although there was a significant reduction of insulin positive cells in female mice in the *A0B2* and *A0B^Δpanc^* groups compared with the *WT* group, there were little overt effects on glucose clearance activity, which is consistent with the description in Gulibaikelamu’s report [13]. Conversely, in male mice, the glucose levels in the *WT* and *A0B2* groups were comparable, which is contradictory to Nishimura’s study [22], which found that *Mafa* is a crucial factor for maintaining mature β-cells. According to a previous study of mice from different background strains, C57BL/6J and ICR, the *Mafa* knockout mice from the C57BL/6J strain exhibited a comparable fasting glucose level compared with the *WT* during the adult stage, however, the phenotype of the mice in the ICR background was more severe. Thus, it is reasonable to infer that variable phenotypes are caused by different strains.

Although various diabetes mouse models have been previously reported, there are several advantages of the *A0B^Δpanc^* mice model. The incidence of male *A0B^Δpanc^* mice is over 90% following a prolonged period of at least 16 weeks. Moreover, the inducible knockout of *Mafb* with *Mafa*^-/-^ background overcame death caused by simultaneous *Mafa* and *Mafb* deletion [11]. Second, owing to the controllable and non-lethal induction of TAM, the *A0B^Δpanc^* mice model could be an efficient candidate for diabetes studies under different physical conditions, preventing death caused by gene manipulation prior to the postnatal stage. Third, the *A0B^Δpanc^* model provides novel insight into the pathogenesis of DM. While, there were no significant differences in CRE and BUN levels, indicating that there may be no obvious kidney dysfunction in *A0B^Δpanc^* mice at least 16 weeks following TAM injection. However, Patel et al. [23] showed that using CRE level to estimate kidney function might be inaccurate. Thus, H&E and MT staining were performed to further detect possible renal lesions in male *A0B^Δpanc^* mice, owing to the abnormal blood and urinary glucose levels in them. Interestingly, male *A0B^Δpanc^* mice showed obvious mesangial matrix expansion and collagen deposition (Figure 5). Taken together, this indicated that while male *A0B^Δpanc^* mice exhibited moderate renal lesions, there were minimal overt effects on renal function. This could be due to two reasons, first, according to previous studies, albuminuria and renal pathological changes are less commonly observed in diabetic C57BL/6J mice than in other strains [24,25,26,27]. Second, the feeding time may also affect underlying pathological changes. Sugimoto et al. [25] have shown that C57BL/6J mice exhibit only mild to moderate changes 6 months following the of induction of DM; a similar phenomenon was reported by Qi et al. [26]. However, in the present study, mice were monitored for 4 months (16 weeks) following the induction of DM. Thus, it is reasonable to infer that the different phenotypes may be caused by different strains and feeding times. In addition, we performed TAM injection on 2-week-old mice to accomplish *Mafb* deletion and accelerate the course of diabetes to possibly induce more severe phenotypes. The fasting glucose levels were detected at 2, 12 and 20 weeks post-TAM, the results showed the fasting glucose levels of both male and female mice in the *A0B^Δpanc^* group were significantly higher than those in the *WT* group at all time points, indicating severely impaired blood glucose tolerance.

*Mafa* and *Mafb*, two fundamental transcription factors play an indispensable role during mouse islet development. The expression patterns are distinct; *Mafa* begins expression at E13.5 in insulin positive cells and persists in β-cells after birth. In contrast, *Mafb* is detected earlier (E10.5) in both insulin-positive and glucagon-positive cells and restricted to α-cells in adulthood [28,29]. In the *A0B^Δpanc^* model, deletion of *Mafb* in β-cells under *Mafa* knockout condition showed more severe phenotypes compared with those in *A0B2* mice, e.g., more impaired glucose tolerance, less/poorly maintained islet structure, and less proportion of insulin positive cells. These are consistent with an earlier study documented by Gulibaikelamu et al. [13], which reported that simultaneous deletion of *Mafa* and *Mafb* aggravates glucose dysfunction and abnormality of islet structure in comparison with *Mafa* single-knockout mice [11]. In addition, our results evidently support Gulibaikelamu’s discovery [13] that *Mafb* has some specific properties for the maintenance of β-cell function, even in adulthood. *Nkx6.1*, a β-cell-restricted transcription factor, is indispensable for maintaining the function of pancreatic β-cells during adulthood [30]. According to the report by Brandon et al. [30], *Nkx6.1* conditional knockout mice showed rapid-onset diabetes by dysregulating biosynthesis and secretion of insulin, and β-cell proliferation. Furthermore, *Nkx6.1* also participates in postnatal β-cell maturation by regulating associated markers [31]. In view of the specific roles of *Mafa* and *Mafb* during β-cell development and maturation, Nkx6.1-Cre driver mouse might be also useful to induce diabetes.

In addition to the persistently high glucose level observed in the *A0B^Δpanc^* mice model, another interesting phenomenon was also notable. The incidence of DM in *A0B^Δpanc^* male mice was markedly higher than that in the *A0B^Δpanc^* female mice. Of note, male mice showed weak Cre recombination activity before TAM injection (Figure 1), which is similar to the results documented by Liu et al. [32], however the reason is unknown. As a result, there may be some *Mafb* deletion before the TAM injection; this could be one possible explanation for the more severe phenotype in male mice. The incidence in other DM models varies widely. For example, in Zucker Diabetic Fatty (ZDF) rats, the females do not develop apparent diabetes [20], while, in BB rats (8 to 16 weeks old), the incidence in both males and females is around 90% [14]. Even one of the most widely used models, NOD mice, also exhibit varying DM incidences based on gender differences [33]; thus implying that gender differences may come from some inherent properties. It has been reported that estrogen can prevent β-cell failure in most common rodent models with β-cell dysfunction, as well as promote β-cell survival [34,35]. Intriguingly, according to Stubbins et al. [36], the male and ovariectomized female C57BL/6J mice showed higher blood glucose level and the abnormality can be rescued by estrogen (E2) treatment.

Inflammation to pancreatic islets is one of the causes of DM. In this study, we did not detect obvious immune cell infiltration in the pancreatic tissues of male *A0B^Δpanc^* mice. However—different from our results—according to Singh et al. [37], loss of *Mafa* and *Mafb* expression promotes inflammation of the pancreatic islets. This may be explained by the following two reasons: First, the mice models used by Singh et al. [37] were *Mafa*^−/−^ and *Mafb^+/−^* mice, which were whole body knockouts, while the mice used by us were *Maf^−/−^*;*Mafb*^flox/flox^, and *Pdx1-CreER*^TM^. Second, the feeding time may also affect the occurrence of some histological changes. Singh et al. [37] kept the mice for 6–8 months, 2-4 months longer than ours. Collectively, owing to the different genotypes and feeding times, the phenotype may be explainable.

In summary, the *A0B^Δpanc^* mice model is a more efficient and stable model for DM research, and is characterized by impaired glucose tolerance, abnormal islet structure, and decreased insulin positive cell number. Owing to the specific functions of MAFA and MAFB, it may therefore be a suitable mediator for detecting the complementary mechanisms between both. On the other hand, there are still some limitations of the *A0B^Δpanc^* model. First, in consideration of the complexity of generation and the availability of *Mafa*^−/−^ and *Mafb*^flox/flox^;*Pdx1-CreER*^TM^ mice, the application of the *A0B^Δpanc^* model might be restricted. Second, a deeper detection may be needed for the metabolic profile, such as lipids metabolism.

## 4. Materials and Methods

### 4.1. Mice

The mouse strain C57BL/6J, used in this study, was derived from Japan SLC (Shizuoka, Japan). The mice were bred under specific-pathogen-free (SPF) conditions with a constant temperature (23.5 ± 2.5 °C) and 12/12 h light-dark cycle, in the Laboratory Animal Resource Center at the University of Tsukuba, Ibaraki, Japan. The procedures were performed in compliance with the guidelines and legal regulations approved by the University of Tsukuba Animal Ethics Committee, Ibaraki, Japan (authorizing No. 19–131, date: 1st June every year).

### 4.2. Generation of A0B^Δpanc^ Mice

*Mafa*-null mice (*Maf^−/−^*, named *A0*) were generated according to the method described by Zhang et al. [8]. *Pdx1*-dependent *Mafb* deletion mice (*Mafb^Δpanc^*) were generated by breeding *Mafb*^flox/flox^ with *Pdx1-CreER*^TM^ transgenic mice (Mutant Mouse Resource & Research Centers (000350-UCD, University of California, Davis, CA, USA)). *Mafb*^flox/flox^ mice with C57BL/6J background were initiated as described in previous reports [38,39]. *Mafa*^−/−^;*Mafb*^flox/flox^;*Pdx1-CreER*^TM^ and *Maf*a^−/−^;*Mafb*^flox/flox^ mice were generated by crossing *Mafa*^−/−^ with either *Mafb*^flox/flox^;*Pdx1-CreER*^TM^ or *Mafb*^flox/flox^, which were designated as *A0B^Δpanc^* and *A0B2* (*B2* indicates *Mafb* wild type), respectively. *WT* mice served as controls.

Induction of the Cre recombination system in the C57BL/6J strain was performed by intraperitoneal (IP) injection of tamoxifen (TAM, Sigma, St. Louis, MO, USA) in 5-week-old mice, following a 5-day period. TAM was mixed with corn oil after dissolving in ethanol and the concentration was 75 mg/kg [40].

### 4.3. Assessment of Cre Activity Induced by Tamoxifen

In mice, efficient Cre reporter strains are potent means to monitor the activity and efficiency of Cre recombinase. Therefore, to evaluate the effectiveness of Cre recombinase, mice were generated by mating *R26GRR* reporter mice (as described by Hasegawa et al. [41]) with *Pdx1-CreER*^TM^ mice. The offspring were identified by molecular characteristic analysis. To confirm the insertion of the Cre reporter, genotype identification was done by PCR. Briefly, the snips of tails were incubated in 50 mM sodium hydrate (NaOH, 600 μL) under 95 ℃ for 30 min, then treated with 1 M tris-hydrochloric acid (Tris-HCl, 50 μL). After that, the samples were subjected to centrifugation (1000 rpm, 2 min) and the supernatant was used for the template. The primers and conditions were as following:

F: 5′-GGACATGTTCAGGGATCGCCAGGCGT-3′

R: 5′-GCATAACCAGTGAAACAGCATTGCTG-3′

95 ℃ 2 min; 95 ℃ 30 s, 60 ℃ 30 s, 72 ℃ 30 s (30 cycles); 72 ℃ 2 min and maintained under 18 ℃. For histological assessment, the heterozygous mice with both *R26GRR* and *Cre* recombinase were implemented with or without TAM injection, respectively. The mice were subjected to isoflurane narcosis, phosphate-buffered saline (PBS) and 4% paraformaldehyde (PFA) perfusion, sequentially. Then, equalized samples were placed in graded sucrose (10, 20 and 30% diluted in PBS). Before cutting, the samples were processed by blocking in Tissue-Tek OCT (Life Technologies, Carlsbad, CA, USA) and freezing in liquid nitrogen. Then, tissues were sectioned with a thickness of 5 μm at −20 ℃. After 20 min incubation at room temperature (25 ± 1 °C), the slides were transferred to −80 ℃ prior to microscopy. The green fluorescent protein (EGFP) and tandem red fluorescent protein (tdsRed) signals were detected by fluorescence microscopy at specific wavelengths.

### 4.4. Intraperitoneal Glucose Tolerance Test (IPGTT)

To evaluate the regulation of β-cells challenged with glucose, IPGTT was used. Prior to IPGTT, mice were fasted for 16 h (17:00–9:00), with free access to water. Then, the mice were subjected to IP injection with a 20% solution of glucose diluted in normal saline (2 g/kg body weight). Subsequently, glucose levels of blood samples from the tail vein were measured at designated time points (0, 15, 30, 60 and 120 min after glucose injection) using a FreeStyle glucometer (Terumo, Tokyo, Japan).

### 4.5. Biochemical Analysis of Blood and Urine

For this assessment, 21-week-old mice (16 weeks post-TAM) were used. Following 24 h in the metabolic cages with normal food and water supply, urine samples were collected and stored at −30 ℃ before use. It was demonstrated that without heating and burning, the integrity of each biochemical component is unaffected. Thus, analysis of urinary glucose was conducted by an automated analyzer (DRI-CHEM 7000V, Fujifilm, Tokyo, Japan). Blood samples from the carotid artery were drawn for BUN and CRE detection from the same mice used for urine sample collection.

### 4.6. Kidney and Pancreas Pathology Analysis

For kidney pathology assessment, 21 week-old *WT*, *A0B2* and *A0B^Δpanc^* male mice (16 weeks post-TAM) were used. The collected tissues were fixed with 4% (*w*/*v*) PFA in 4 ℃, then embedded in paraffin for further analysis. The paraffin-embedded kidneys were serially sectioned with 2 μm thickness, and stained with hematoxylin–eosin (H&E) and Masson trichrome (MT) for detection of cell structure changes and collagen matrix deposition. For pancreas pathology analysis, both male and female mice at 21 weeks old in each group were used after the same treatment as kidney samples. The sectioned slices (2 μm) of pancreases were subjected to H&E staining to detect possible pathological changes.

### 4.7. IHC Analysis and Cell Counting

The 21-week-old *WT*, *A0B2* and *A0B^Δpanc^* mice (16 weeks post-TAM) were subjected to pancreatic tissue collection. After collection, the samples were rapidly fixed in 4% (*w*/*v*) PFA under 4 ℃ incubation, then embedded in paraffin. Sectioned 2 μm slices were processed through xylene for deparaffinization, 3 times, 10 min each, then treated with graded ethanol for rehydration. Before blocking, the sections were washed with tap water (5 min) and PBS (3 times, 5 min each). Afterwards, immunostaining was conducted. Prior to treating with primary antibody, the sections were blocked by submersion within PBS containing 10% goat serum for 1 h at 25 ℃. Subsequently, the primary antibodies, guinea pig anti-insulin (ab7842, Abcam, Cambridge, UK) and rabbit anti-glucagon (2760S, Cell Signaling Technology, Danvers, MA, USA), were used for processing sections at ratios of 1:100 and 1:500, respectively. After an overnight incubation at 4 ℃, species-matched Alexa Fluor-conjugated secondary antibodies (A11076, goat anti-guinea pig antibody-Alexa Fluor 594 and A11034, goat anti-rabbit antibody-Alexa Fluor 488, Life Technologies, Carlsbad, CA, USA) were added before 1 h incubation at 25 ℃. For further visualization, all the images were subjected to fluorescence microscopy (Biorevo X-8000, Keyence, Osaka, Japan) after PBS washing (3 times, 5 min each).

To identify the characteristics of impaired islets, after immunofluorescence staining, the specific hormone positive cells in each islet were counted using islet microscopy images, manually. Representative islets (15–25) from 2–3 mice in each group were counted. The number of each hormone specific cell was manually detected using ImageJ software (National Institutes of Health, Bethesda, MD, USA), to calculate their proportions within the islets; the figures were divided by the total Hoechst positive number from the same islets.

### 4.8. Statistical Analysis

All the experimental data are exhibited as means ± standard deviations (SD). Statistical analysis between two groups were performed using the Student’s *t*-test, with multiple biological replicates (≥2), all the *P* values were adjusted by Bonferroni Correction, which was performed by R and the differences among the groups were considered significant at adjusted *p* values <0.05.

## Figures and Tables

**Figure 1 ijms-21-05606-f001:**
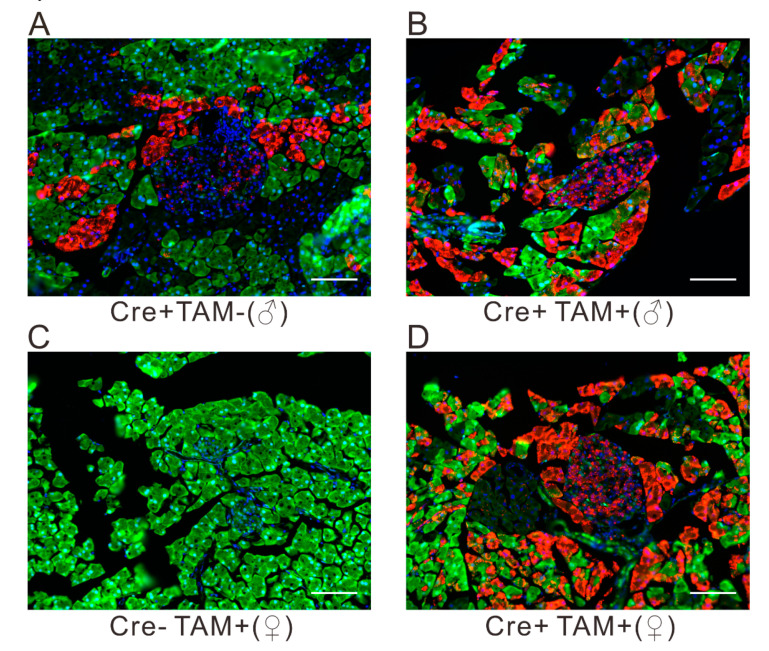
Cre-mediated enhanced green fluorescent protein (EGFP) and tdsRed expression in *R26GRR/Pdx1-CreER*^TM^ adult mice, without (**A**) or with (**B**,**D**) TAM administration. The *Pdx1-CreER*^TM^ (−) mice (**C**) or the mice without (**A**) TAM injection are shown as negative controls. Scale bar, 50 μm.

**Figure 2 ijms-21-05606-f002:**
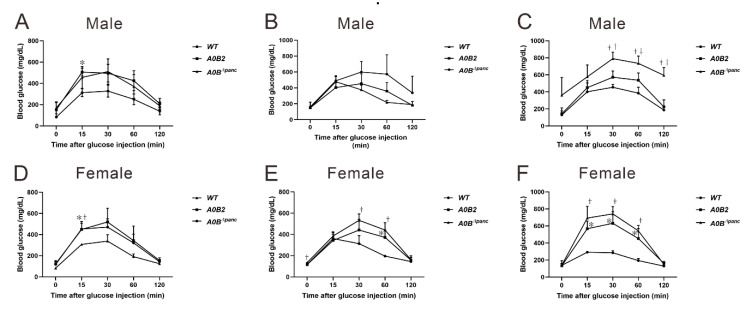
Effect of *Mafb* deletion on glucose clearance activity in *A0B^Δpanc^* mice. The intraperitoneal glucose tolerance test (IPGTT) was performed at designated timepoints (0, 4, and 16 week post-TAM) between male (**A**–**C**) and female (**D**–**F**) mice of indicated genotypes. Prior to IPGTT, 2 g/kg glucose was intraperitoneally injected after a 16 h period starvation. The data are from 3–7 mice of indicated genotypes. * *A0B2* vs. *WT*, *p* < 0.05; † *A0B^Δpanc^* vs. *WT*, *p* < 0.05; ‡ *A0B2* vs. *A0B^Δpanc^*, *p* < 0.05. The error bars indicate SD.

**Figure 3 ijms-21-05606-f003:**
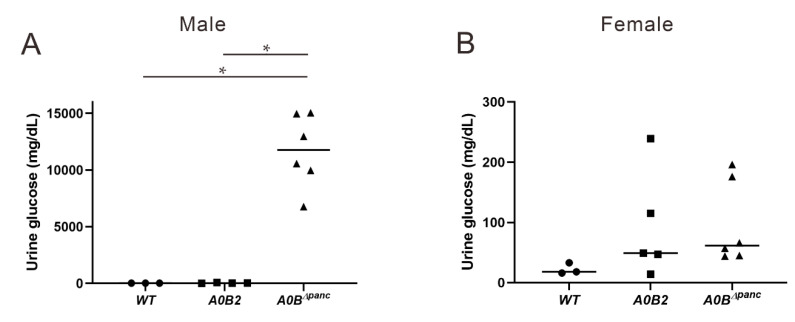
Urine glucose changes among *WT*, *A0B2* and *A0B^Δpanc^* mice. Following 24 h urine collection, glucose level was assayed on 16-week post-TAM male (**A**) and female (**B**) mice of indicated genotypes. The data come from 3–6 mice of each genotype, * *p* < 0.01.

**Figure 4 ijms-21-05606-f004:**
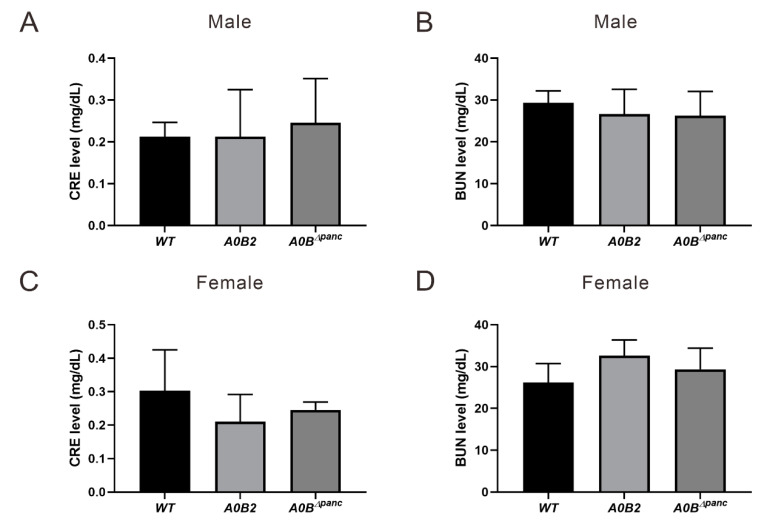
Metabolic parameters analysis. Blood samples drawn from the carotid artery were performed to analyze induced renal dysfunction evidenced by blood urea nitrogen (BUN) and serum creatinine (CRE) analysis at 16 weeks post-TAM injection, (**A**,**B**) male, (**C**,**D**) female. About 3–6 mice in each group were used. The error bars indicate SD.

**Figure 5 ijms-21-05606-f005:**
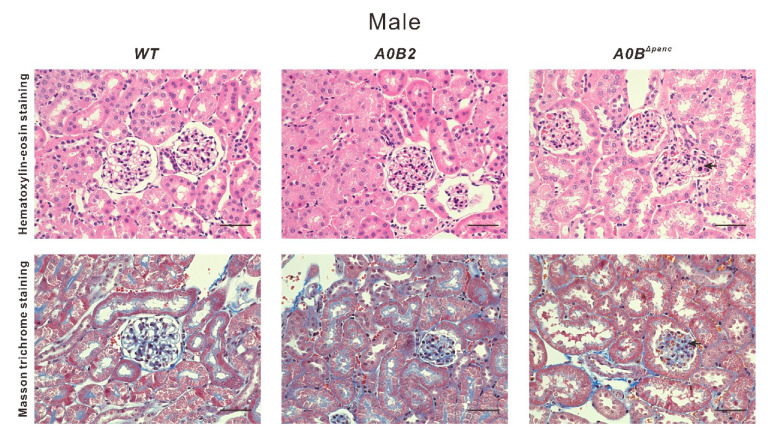
Morphologic comparison of kidneys from male *WT*, *A0B2* and *A0B^Δpanc^* mice were performed via staining with hematoxylin–eosin and Masson trichrome. Representative glomeruli are from mice of indicated genotypes at 21 weeks old. Arrows indicate expansion of mesangial matrix and increased collagen in the mesangial matrix, respectively. Scale bar, 50 µm.

**Figure 6 ijms-21-05606-f006:**
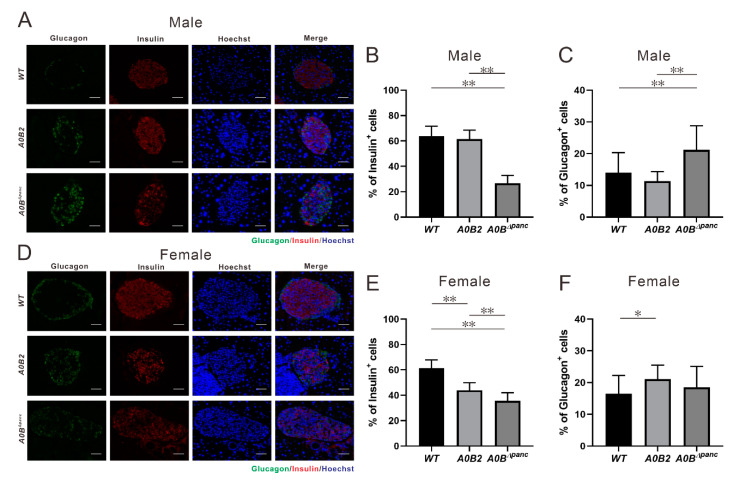
Immunohistochemistry analysis and specific hormone cell number counting, (**A**–**C**, male; **D**–**F**, female). (**A**,**D**) Immuno-staining of glucagon (green) and insulin (red) in pancreatic islets from 16 weeks post-TAM injection in mice of each indicated genotype. (**B**,**E**) Insulin positive cell number/total islet cell number in pancreatic islets of each group. The results are from 15–25 islets of 2–3 mice in each genotype. (**C**,**F**) Glucagon positive cell number/total islet cell number in pancreatic islets of each group. * *p* < 0.05; ** *p* < 0.01, the error bars indicate SD. Scale bar, 50 μm.

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
