# Peer review of "An Inducible Diabetes Mellitus Murine Model Based on MafB Conditional Knockout under MafA-Deficient Condition"

_ijms, 2020, doi:10.3390/ijms21165606_

Round 1

Reviewer 1 Report

In revised manuscript, the authors appear to have adequately addressed all major and minor comments

Author Response

We appreciate taking the time to review our article.

Reviewer 2 Report

This is interesting and well-written paper. I have only some minor comments to the Authors.

  1. Did the Authors observed differences in body mass between wt, A0B2 and A0BΔpanc animals? Furthermore, it is disappointing that metabolic profile (e.g. lipids) of this new diabetic model was not deeply analyzed.
  2. Can Authors explain why circulating insulin levels and insulin tolerance tests were not assessed in this work?
  3. no. of antibodies used in this work should be provided.
  4. Description of PCR analysis is missing.
  5. The Authors claim that they did not find any inflammation markers in endocrine pancreas. This point should be descripted in more detailed-fashion in materials and methods. In addition, I believe that it would be worth to add these results to the manuscript.
  6. It the discussion the Authors should list and discuss the limitation of this study.

Author Response

Dear Editors and Reviewers:

Thank you for your letter and the reviewers’ comments concerning our manuscript entitled “An inducible diabetes mellitus murine model based on MafB conditional knockout under MafA-deficient condition” (Manuscript ID: ijms-867172). Those comments are all valuable and very helpful for revising and improving our paper, as well as important guiding significance to our research. We have studied comments carefully and made corrections which we hope meet up with approval. The main corrections in the paper and the responses to the reviewer’s comments are as following:

(For your convenience, please change the options of Track Changes in the manuscript as follows: Track ChangesAll MarkupMarkup optionsBalloonsShow All Revisions Inline

Reviewer 2

This is interesting and well-written paper. I have only some minor comments to the Authors.

  1. Did the Authors observed differences in body mass between wt, A0B2 and A0BΔpanc animals? Furthermore, it is disappointing that metabolic profile (e.g. lipids) of this new diabetic model was not deeply analyzed.

Response: Thank you for your comments. The body weights of the mice in WT, A0B2 and A0BΔpanc group were recorded from 4-weeks to 16-weeks post-TAM (Figure 1). Obviously, the body mass of A0BΔpanc male mice was significantly lower than that in WT group at all time points, while, there was no significant difference between A0B2 and A0BΔpanc group. In contrast, the body mass of female mice in WT and A0BΔpanc group was comparable. Furthermore, about the metabolic profile (e.g. lipids), we really appreciate your kind suggestion, it is true that the possible changes of metabolic profile will enrich the phenotypes of our model. We will carry out the associated analysis about that in the following study. The associated description of body mass has been added in the manuscript (page 3, lines 108-110).

Figure 1. Body mass of both male and female mice in WT, A0B2 and A0BΔpanc group at designated time points (4-, 8-, 12- and 16-weeks post-TAM). The data are from 3-7 mice of each genotype. *: A0BΔpanc vs WT, P <0.05; *: A0BΔpanc vs A0B2, P <0.05; #: A0B2 vs WT, P <0.05, The error bars indicate SD.

  1. Can Authors explain why circulating insulin levels and insulin tolerance tests were not assessed in this work?

Response: Thank you for your rigorous comments. Since most of our results were consistent with the symptoms of Type 1 Diabetes, such as high urine and blood glucose levels, decreased insulin positive cell number, as well as losing body mass. Thus, we may pay a little attention on the contents you mentioned. We are still carrying on this project (TAM injection performed at 2-week-old mice), and we will try to analyze the circulating insulin levels and insulin tolerance to enrich the phenotype of our model.

  1. of antibodies used in this work should be provided.

Response: Thank you for your kind suggestion. The catalog number of each antibody has been added in the manuscript (page 11, lines 377-381).

  1. Description of PCR analysis is missing.

Response: Sorry for the missing description about PCR analysis. As you suggested, the details about PCR analysis (method and result) have been added in the manuscript as following:

Page 10, lines 329-337, “To confirm the insertion of Cre reporter, genotype identification was done by PCR. Briefly, the snips of tails were incubated in 50 mM sodium hydrate (NaOH, 600 μl) under 95℃ for 30 min, then treated with 1 M tris-hydrochloric acid (Tris-HCl, 50 μl). After that, the samples were subjected to centrifugation (1000 rpm, 2 min) and the supernatant was used for template. The primers and conditions were as following:

F:5′-GGACATGTTCAGGGATCGCCAGGCGT-3′,

R: 5′-GCATAACCAGTGAAACAGCATTGCTG-3′.

95℃ 2 min; 95℃ 30s, 60℃ 30s, 72℃ 30s (30 cycles); 72℃ 2 min and maintained under 18℃.

Page 3, line 98, “The genotype of the progeny was confirmed by polymerase chain reaction (PCR) with single and bright bands were observed (data not shown).”

  1. The Authors claim that they did not find any inflammation markers in endocrine pancreas. This point should be descripted in more detailed-fashion in materials and methods. In addition, I believe that it would be worth to add these results to the manuscript.

Response: Thank you for your rigorous comments. As you suggested, the description about materials and methods have been added (Page 11, lines 360-368) the details are as following:

“For pancreas pathology analysis, both male and female mice at 21-week-old in each group were used. After the same treatment as kidney samples. The sectioned slices (2-µm) of pancreases were subjected to Hematoxylin-eosin (H&E) staining to detect possible pathological changes.”

On the other hand, to detect the possible pancreatic inflammation, a total of 15 slides stained with Hematoxylin-eosin were analyzed (Male: WT, n=2; A0B2, n=2; A0BΔpanc, n=3; Female: WT, n=2; A0B2, n=3; A0BΔpanc, n=3). Unfortunately, there were no inflammation marker cells observed, such as lymphocyte and neutrophil. Thus, there may no obvious pancreatic inflammation occurred in our model at 16-weeks post TAM injection, theoretically. Although it is different with the report of Singh et al [1], we furtherly discussed the possible reasons in the discussion part, which may be explained by different genotypes and feeding times. We really appreciate your precise comments, and we will perform a more comprehensive analysis in the following study.

  1. It the discussion the Authors should list and discuss the limitation of this study.

Response: Thank you for your kind advice. It is true that the summary and discussion of the limitation may be more intuitive and easier for audience. As you suggested, we have listed and discussed more about the limitation of our model at the end of discussion (Page 9, lines 299-303). The details are as following:

“On the other hand, there are still some limitations. First, in consideration about the complexity of generation and the availability of Mafa-/- and Mafbflox/flox;Pdx1-CreERTM mice, the application of A0BΔpanc model might be restricted. Second, a deeper detection may be needed for metabolic profile, such as lipids metabolism.”

References

  1. Singh, T.; Colberg, J. K.; Sarmiento, L.; Chaves, P.; Hansen, L.; Bsharat, S.; Cataldo, L. R.; Dudenhöffer-Pfeifer, M.; Fex, M.; Bryder, D.; Holmberg, D.; Sitnicka, E.; Cilio, C.; Prasad, R. B.; Artner, I. Loss of MafA and MafB expression promotes islet inflammation. Rep. 2019, 9, 9074.

*This manuscript is a resubmission of an earlier submission. The following is a list of the peer review reports and author responses from that submission.*

Round 1

Reviewer 1 Report

Deng et al describe a novel experimental model of Diabetes in mice. This has been achieved via Pdx1-dependent Mafb-deletion mice under Mafa knockout condition (A0BΔpanc), by using tamoxifen-inducible Cre-loxP system.

Data are interesting. However, several major concern may be raised on many aspects related to the validation of this model in order to be consistent and useful in translational studies:

1) The model seems to correlate with Type 1 DM since impaired islet structure  and a decreased proportion of insulin positive cell were observed. However, no fasting glucose changes were found.

2) The impaired glucose tolerance, which suggests the induction of DM, is not accompanied by micro-angiopathic lesions in the kidney and this makes the model unable to cover vascular and renal alterations accompanying DM very early in the course of the disease

3) No changes in cytokines and adipokynes are described to verify whether or not the experimental model may be useful in studying inflammation which occurs in DM

4) Gender differences are not properly explained

In my view, the model provides very contradictory informations and seems do not to be reliable for providing evidence on physio-pathology and treatment of DM useful to be translated into human studies

Reviewer 2 Report

Pancreatic and duodenal homeobox 1 (Pdx1) is a homeodomain-containing transcription factor specific for pancreatic islet β cells. Murine MafA and MafB are 2 islet-enriched transcription factors that can influence transcriptions of those genes associated with glucose sensing and hormone secretion, and are highly homologous to their respective human orthologs. In this manuscript, the authors created Pdx1-dependent Mafb-deletion mice under Mafa knockout condition (A0BΔpanc) to induce a murine model for diabetes mellitus (DM) at a specific time point, by applying tamoxifen-inducible Cre-loxP system. After 16 weeks, metabolic phenotypes were characterized by performing intraperitoneal glucose tolerance test (IPGTT), urine glucose test, and metabolic parameters analysis. Male A0BΔpanc mice were observed to exhibit obvious impaired glucose tolerance, and high urine glucose level. Also, impaired islet structure and decreased proportion of insulin positive cell were identified. Taken together, the A0BΔpanc murine model created by the authors could provide an efficient inducible model for DM research.  

(I) Major Comments Type 2 diabetes (T2D) is a metabolic disease resulting from multiple genetic factors that can lead to remarkable complications in a variety of tissues. Rodent models, and in particular, mouse models, have been used widely for studying pathophysiology underpinning T2D (e.g., Neubauer N, Kulkarni RN. Molecular approaches to study control of glucose homeostasis. ILAR J. 2006;47:199-211. PMID: 16804195). Traditional "global" knock-out mouse model has significant limitations of possibly causing inviability at embryonic stage when essential genes are constitutively deleted, and the Cre-loxP technology combined with tamoxifen-inducible system allows conditional deletion of target gene(s) via the Cre-loxP system to attain temporally and spatially controllable gene expression, which can surmount those limitations of conventional "global" knock-out mouse models (e.g., Valny M, Honsa P, Kirdajova D, Kamenik Z, Anderova M. Tamoxifen in the Mouse Brain: Implications for Fate-Mapping Studies Using the Tamoxifen-Inducible Cre-loxP System. Front Cell Neurosci. 2016;10:243. PMID: 27812322.). Generally speaking, the experiments appear to be appropriately conducted and the manuscript is clearly written.   I have the following major comments.  

(1) Page 2, 2nd paragraph, lines 59-60, The authors stated that"Among them, MAFA and MAFB, two counterparts exhibiting different expression patterns have attracted attention [5]" Reference [5] refers to Hang Y, Stein R. MafA and MafB activity in pancreatic β cells. Trends Endocrinol Metab. 2011;22:364-73. PMID: 21719305. However, the authors could add another recent review paper on the biological functions of Maf protein family members to support this statement, e.g., Tsuchiya M, Misaka R, Nitta K, Tsuchiya K. Transcriptional factors, Mafs and their biological roles. World J Diabetes. 2015;6:175-83. PMID: 25685288; PMCID: PMC4317310.  

(2) Page 2, 3rd paragraph, lines 72-73,
The authors stated that "Animal modeling, an efficient approach for exploring and characterizing disease etiology and pathophysiology, also provides novel insights into the development of therapeutic treatments."
The authors could add appropriate references to support their statement. E.g., (i) Mullen Y. Development of the Nonobese Diabetic Mouse and Contribution of Animal Models for Understanding Type 1 Diabetes. Pancreas. 2017;46:455-466. PMID: 28291161;(ii) Cefalu WT. Animal models of type 2 diabetes: clinical presentation and pathophysiological relevance to the human condition. ILAR J. 2006;47:186-98. PMID: 16804194; and(iii) Gurumurthy CB, Lloyd KCK. Generating mouse models for biomedical research: technological advances. Dis Model Mech. 2019;12:dmm029462. PMID: 30626588.

(3) Page 4, Figure 2, Although Figure 2's legend described that panels (A)-(C) are for male mice, and panels (D)-(F) are for female mice, respectively, it would be better to clearly label each of panels (A), (B) and (C) with "Male" on the top of each panel, and each of panels (D), (E) and (F) with "Female" on the top of each panel, respectively. So that the audience can readily distinguish the difference.   (4) Page 4, lines 136-138,the authors stated that "there was no considerable difference between these two groups (A0BΔpanc: 11707 ± 2947 mg/dL; A0B2: 33.75 ± 23.27 mg/dL; WT: 23.67 ± 3.30 mg/dL) (Figure 3A)" However, the authors shall explicitly present the Bonferroni-corrected P-values for the 3 comparisons, i.e., A0BΔpanc vs A0B2, A0BΔpanc vs WT, and A0B2 vs WT, respectively.  

(5) Page 5, Figure 3, Although Figure 3's legend described that panel (A) is for male mice, and panel (B) is for female mice, respectively, it would be better to clearly label panel (A) with "Male" on the top of the panel, and label panel (B) with "Female" on the top of the panel, respectively. So that the audience can readily distinguish the difference.  

(6) Page 5, Figure 4, Although Figure 4's legend described that panels (A) and (B) are for male mice, and panels (C) and (D) are for female mice, respectively, it would be better to clearly label each of panels (A) and (B) with "Male" on the top of the panel, and label each of panels (C) and (D) with "Female" on the top of the panel, respectively. So that the audience can readily distinguish the difference.  

(7) Page 6, Figure 5, Although Figure 5's legend described that panels (A)-(C) are for male mice, and panels (D)-(F) are for female mice, respectively, it would be better to clearly label each of panels (A), (B), and (C) with "Male" on the top of the panel, and label each panels (D), (E), and (F) with "Female" on the top of the panel, respectively. So that the audience can readily distinguish the difference.  

(8) Page 7, lines 203-204, the authors stated that "which is contradictory to Nishimura’s study, which found that MAFA is a crucial factor for maintaining mature β-cells" The authors shall add the reference ID for "Nishimura’s study" [which is reference (17) in the manuscript], and shall change "MAFA" to "Mafa" in the above statement.  

(9) Page 7, lines 215-216, the authors stated that  "Third, the A0BΔpanc model provides novel insight into the pathogenesis of DM since most animal models have little relevance to human diabetes."
The expression "most animal models have little relevance to human diabetes" is not very appropriate, and I would suggest the authors revise the above statement to "Third, the A0BΔpanc model provides novel insight into the pathogenesis of DM."

(10) Page 7, lines 232-234, the authors stated that
"our results evidently support the Gulibaikelamu’s discovery, MAFB has some specific properties for the maintenance of β-cell function even in adulthood" The authors shall add the reference ID for " Gulibaikelamu’s discovery" [which is reference (11) in the manuscript], and shall change "MAFB" to "Mafb" in the above statement.

(11) Page 9, lines 323-325, the authors stated that  "Statistical analysis between two groups were performed using the Student’s t-test, with multiple biological replicates (≥3), the P values were adjusted by Bonferroni Correction" “multiple biological replicates (≥3)” should be corrected to “multiple biological replicates (≥2)” (e.g., Figure 5's legend states that "The results are from 15-25 islets of 2-3 mice in each genotype"), and in each of Figure 2, Figure 3, Figure 4, and Figure 5’s respective panels, how many statistical tests were conducted and therefore were adjusted by applying the Bonferroni Correction [for conducting n statistical tests, the Bonferroni-corrected P-value is n*(nominal P-value)]. Are all the P-values presented for Figures 2, 3, 4 and 5, Bonferroni-corrected P-values, or unadjusted P-values (i.e., nominal P-values)? Please clearly state the respective P-values for the respective panels of these figures, and in the main text of the manuscript, so that the audience can unequivocally understand whether the P-values presented are Bonferroni-corrected P-value or not. Further, what statistical software computer program was applied (e.g., SAS, R, or GraphPad Prism)? The authors shall provide the name of the computer program and the version of such program used for performing statistical analyses.  

(12) In this paper, the authors applied the Cre/Lox recombination-mediated conditional gene knockout technology to study gene functions in a specific tissue, i.e., pancreas, and an inducible Cre recombinase was apply to inactivate a target gene (i.e., Mafb) at a certain time point, and the authors generated Pdx1-dependent Mafb deletion mice (MafbΔpanc) by breeding Mafbflox/flox with Pdx1-CreERTM transgenic mice. Nkx6.1, a homeodomain transcription factor, also play a critical role in pancreatic islet beta-cell development (e.g., Schisler JC, et al., Mol Cell Biol. 2008;28:3465-76. PMID: 18347054), such that Nkx6.1 mutant mice showed a reduction of pancreatic beta-cells (e.g., Sander M, et al., Development. 2000;127:5533-40. PMID: 11076772). A study of Memon B, et al. (Stem Cell Res Ther. 2018;9:15. PMID: 29361979) suggested that both Padx1 and Nkx6.1 are the indispensable precursors of functional pancreatic beta cells. A limitation of the current study is that only Pdx1-dependent Mafb deletion mice were generated, but Nkx6.1 was not studied by their approach. The authors shall add this point in the "3. Discussion" section.  

(II) Minor Comments There are several typographical and grammatical errors that should be corrected, which are shown in the following:  

Page 1, line 33,
"are highly homologous to human protein ortholog" could be changed to "are highly homologous to human protein orthologs"

Page 1, lines 38-39,
"impaired islet structure and decreased proportion of insulin positive cell was observed" could be changed to "impaired islet structure and decreased proportion of insulin positive cells were observed"

Page 2, line 57,
"Increasing evidence have demonstrated that islet-enriched transcription factors" could be changed to
"An increasing amount of evidence has demonstrated that islet-enriched transcription factors"

Page 2, line 60,
"in rodents, MAFB starts its transient production" could be changed to "in rodents, Mafb starts its transient production"

Page 2, lines 62-63,
"In contrast, MAFA expression in β-cells initiates at E13.5 and its function persist into adulthood [6, 7]. Many reports have uncovered the specific roles of MAFA" could be changed to "In contrast, Mafa expression in β-cells initiates at E13.5 and its function persists into adulthood [6, 7]. Many reports have uncovered the specific roles of Mafa"

Page 2, line 65,
"In contrast, Mafb knockout mice was restricted" could be changed to "In contrast, Mafb knockout mice were restricted"

Page 2, line 75,
"non-obese diabetic (NOD) mice, Akita mice, and BB rats" could be changed to
"non-obese diabetic (NOD) mice, Akita mice, and Bio-Breeding (BB) rats"

Page 3, line 105,
"IPGTT was implemented to assess the ability" could be changed to
" Intraperitoneal glucose tolerance test (IPGTT) was implemented to assess the ability"

Page 3, line 109,
"at 30 min was significant higher in" could be changed to "at 30 min was significantly higher in"

Page 3, line 114,
"at all time-points during the following 90 min" could be changed to "at all time points during the following 90 min"

Page 3, line 114,
"at all time-points during the following 90 min" could be changed to "at all time points during the following 90 min"

Page 4, lines 117-118,
"the blood glucose levels showed a similar pattern at 4- and 16-weeks" could be changed to "the blood glucose levels showed similar patterns at 4- and 16-weeks"

Page 4, lines 119-120,
"decreased to normal levels gradually (Figure 2E and F)" could be changed to "decreased to normal levels gradually (Figure 2E and 2F)"

Page 5, line 163,
"Furthermore, the number and percentages of insulin and glucagon positive cells" could be changed to "Furthermore, the numbers and percentages of insulin and glucagon positive cells"

Page 7, line 198,
"both male and female mice (Figure 5 A-B, D-E). Intriguingly" could be changed to "both male and female mice (Figure 5A-5B, 5D-5E). Intriguingly"

Page 7, lines 202-203,
"the glucose levels in the WT and A0B2 group were comparable" could be changed to "the glucose levels in the WT and A0B2 groups were comparable"

Page 7, lines 223-225,
"MAFA and MAFB, two fundamental transcription factors play an indispensable role during mouse islet development. While, the expression patterns are distinct, MAFA begins expression at E13.5 in insulin positive cells and persists in β-cells after birth. In contrast, MAFB is detected" could be changed to "Mafa and Mafb, two fundamental transcription factors play an indispensable role during mouse islet development. While, the expression patterns are distinct, Mafa begins expression at E13.5 in insulin positive cells and persists in β-cells after birth. In contrast, Mafb is detected"

Page 8, line 241,
"For example, in ZDF rats, the females do not develop apparent diabetes [15]" could be changed to "For example, in Zucker Diabetic Fatty (ZDF) rats, the females do not develop apparent diabetes [15]"